# Rapid and Simple Detection of *Burkholderia gladioli* in Food Matrices Using RPA-CRISPR/Cas12a Method

**DOI:** 10.3390/foods12091760

**Published:** 2023-04-24

**Authors:** Jiale Zheng, Li Liu, Xiangmei Li, Zhenlin Xu, Zuoqi Gai, Xu Zhang, Hongtao Lei, Xing Shen

**Affiliations:** 1Guangdong Provincial Key Laboratory of Food Quality and Safety, National-Local Joint Engineering Research Center for Machining and Safety of Livestock and Poultry Products, South China Agricultural University, Guangzhou 510642, China; zhengjl127@163.com (J.Z.); m19865040105@163.com (L.L.); lixiangmei12@163.com (X.L.); jallent@163.com (Z.X.);; 2Guangzhou Editgene Co., Ltd., Guangzhou 510630, China; 3Guangdong Laboratory for Lingnan Modern Agriculture, Guangzhou 510642, China

**Keywords:** food safety, *Burkholderia gladioli*, foodborne pathogen bacteria, CRISPR/Cas12a, rapid detection

## Abstract

Pathogenic variants of *Burkholderia gladioli* pose a serious threat to human health and food safety, but there is a lack of rapid and sensitive field detection methods for *Burkholderia gladioli*. In this study, the CRISPR/Cas12a system combined with recombinant enzyme polymerase amplification (RPA) was used to detect *Burkholderia gladioli* in food. The optimized RPA-CRISPR/Cas12a assay was able to specifically and stably detect *Burkholderia gladioli* at a constant 37 °C without the assistance of large equipment. The detection limit of the method was evaluated at two aspects, the genomic DNA (gDNA) level and bacterial quantity, of which there were 10^−3^ ng/μL and 10^1^ CFU/mL, respectively. Three kinds of real food samples were tested. The detection limit for rice noodles, fresh white noodles, and glutinous rice flour samples was 10^1^ CFU/mL, 10^2^ CFU/mL, and 10^2^ CFU/mL, respectively, without any enrichment steps. The whole detection process, including sample pretreatment and DNA extraction, did not exceed one hour. Compared with the qPCR method, the established RPA-CRISPR /Cas12a method was simpler and even more sensitive. Using this method, a visual detection of *Burkholderia gladioli* that is suitable for field detection can be achieved quickly and easily.

## 1. Introduction

*Burkholderia gladioli*, a kind of gram-negative bacterium, is widely distributed in nature and have been isolated from plants, the environment, and food [1,2,3,4,5]. It has interspecies diversity, including four pathogenic types [6,7], *Burkholderia gladioli* pv. *gladioli, Burkholderia gladioli* pv. *alliicola*, *Burkholderia gladioli* pv. *agaricicola*, and *Burkholderia gladioli* pv. *cocovenenans*. The first three types were phytopathogenic strains, which could cause rot symptoms in important cash crops such as rice [8], gladiolus [6], and onions and in expensive ornamental plants, such as cymbidium [9,10]. Among all types, *Burkholderia gladioli* pv. *cocovenenans* (*B. cocovenenans*) is a kind of foodborne pathogen first found in Asian countries [11]. It can produce a kind of deadly toxin named bongkrekic acid, which is the main reason for food poisoning and death caused by *B. cocovenenans* [12]. Fermented cereal products, spoiled tremella fungus, and potato products are the main food sources of pollution [13]. Food poisoning events caused by *B. cocovenenans* occurred mainly in China, Indonesia, and Mozambique [14] while the vast majority of the reports were from China. According to the statistics, food poisoning cases caused by this bacterium have occurred in at least 16 provinces in China, and there is currently no specific treatment for it [15]. *B. cocovenenans* is the foodborne pathogen with the highest morbidity and mortality found so far in China [11]. Death caused by *B. cocovenenans* has been reported from time to time. The most famous event occurred in 2020; all nine people in one family died of acid soup poisoning during a family dinner in Heilongjiang province with a mortality rate of 100% [16]. It has caused serious threats to human health and food safety. In view of these problems, timely detection and control of the contamination of *Burkholderia gladioli* in food is urgently needed.

The detection of *Burkholderia gladioli* mainly focuses on the individual detection of some pathogenic subspecies [17,18,19], and these detection methods provide precise identification for subspecies. However, as some of the phytopathogenic subspecies could also cause human diseases [20,21], universal detection methods for all *Burkholderia gladioli* strains are necessary. At present, a series of methods have been developed for the detection of *Burkholderia gladioli*, including PCR [22,23,24], PCR-RFLP [25], qPCR [26], biochemical test [27], DNA microarray [28], dPCR, and MALDI-TOF-MS [29,30]. However, these methods are based on expensive equipment, which are time-consuming or complex and limited to laboratorial detection. Hence, it is necessary to establish a more convenient, simpler, and more sensitive detection method to meet the needs of field detection.

The Clustered Regularly Interspaced Short Palindromic Repeats (CRISPR) system and CRISPR-associated proteins (Cas) are known for their powerful gene editing capabilities [31,32]. In recent years, the particular sequence recognition capability of the CRISPR/Cas system and the non-specific cleavage activity of Class II Cas proteins also demonstrate the rapid, ultra-high specificity and ultra-sensitivity for nucleic acid target recognition combined with the amplification technologies [33,34,35]. In 2017, the Feng Zhang team combined Cas13a with RPA (Recombinase polymerase amplification) technology, which was called the Specific High-Sensitivity Enzymatic Reporter UnLOCKing (SHERLOCK) platform for RNA target recognition and detection, that realized the detection of single nucleic acid molecules [35]. In 2018, the Cas12a-based detection technology DNA Endonuclease Targeted CRISPR Trans Reporter (DETECTR) and One-hour low-cost Multipurpose Highly Efficient systems (HOLMES) were set up to achieve aM level DNA detection [36,37]. These technologies mature rapidly and are widely used in clinical diagnosis [38,39], pathogenic microorganism identification [40,41], genetically modified crop detection [42], and other rapid nucleic acid detection scenarios. The detection technology based on the CRISPR/Cas system has already been developed into detection products and applied to practice in medical diagnosis. However, the translation of this technology into practical applications in food safety testing is more challenging due to the great interference brought by complex food matrices.

In this study, technologies of CRISPR/Cas12a combined with RPA were employed to establish a rapid detection method for *Burkholderia gladioli* based on a conservative mitochondrial gene. Without using large instruments and a complex electrophoresis analysis, the established RPA-CRISPR/Cas12a method can be carried out easily in a very short time and has proven to be useful for the rapid and sensitive detection of *Burkholderia gladioli* in rice noodles, fresh white noodles, and glutinous rice flour. It can provide a technical reference for the rapid field detection of *Burkholderia gladioli* and can promote the development of detection technologies in food safety monitoring.

## 2. Materials and Methods

### 2.1. Materials and Reagents

*Burkholderia gladioli* (CICC10574) were purchased from Yuweitech Technology (Beijing, China); *Burkholderia gladioli* (ATCC10248), *Burkholderia gladioli* pv. *cocovenenans*, *Salmonella enteritis* (CMCC50041), *Vibrio parahaemolyticus*, *Listeria monocytogenes*, *Bacillus cereus*, *Staphylococcus aureus* were all from the Laboratory of Food Quality and Safety (Guangzhou, China). TwistAmp Liquid Basic kit was purchased from TwistDx Ltd. (Cambridge, UK). TIANamp Bacteria DNA Kit was purchased from Tiangen Biochemical Technology (Beijing, China). Cas12a enzyme was provided by Editgene Co., Ltd. (Guangzhou, China). The crRNA and ssDNA-FQ reporting probes (FAM-TTTTTT-BHQ) were synthesized by Genewiz Biotechnology (Suzhou, China). RNase inhibitors were purchased from Xinhai Gene Testing. NEBuffer2.1 was purchased from New England Biolabs Inc. (Ipswich, MA, USA). Rice noodles, fresh white noodles, and glutinous rice flour samples were purchased from the Triangle Market in the South China Agricultural University (Guangzhou, China). 2 × TaqMan Fast qPCR Master Mix was purchased from Bioengineering (Shanghai, China).

### 2.2. DNA Extraction

The dry powder strain or glycerol strain was cultured in liquid medium at 37 °C and centrifuged at 250 rpm for 24 h to obtain the bacterial cells in logarithmic growth phase. In order to get better activity of the strain and ensure no contamination, single colony was picked and cultured in liquid medium again. The obtained bacterial solution was centrifuged at 10,000 rpm for 1min, and the precipitate was collected for DNA extraction. In order to obtain higher purity and concentration, DNA was extracted with the TIANamp bacterial DNA extraction kit following the kit’s instructions. 

### 2.3. Screening of Primers and Establishment of RPA Amplification System

The 16S-23S rRNA intergenic region sequences of bacteria are usually used to identify the genetic relationships among bacteria [43]. The highly conserved 16S-23S rRNA sequence in *Burkholderia gladioli* (accession ID EF552059) was selected as the target for the design of specific RPA amplification primers. According to the RPA primer design principles, a total of nine pairs of primers were designed using SnapGene and Primer Premier 5.0 software (Appendix A). In order to test the specificity of the primers, genomic DNA (gDNA) extracted from *Burkholderia gladioli*, *Salmonella*, *Vibrio parahemolyticus*, *Escherichia coli*, *Listeria monocytogenes Bacillus cereus*, and *Staphylococcus aureus* was used for RPA amplification. The initial reaction system, including 0.25 μM forward/reverse primer, 14 mM MgOAc, 1.6 mM dNTPs, and 1 μL gDNA, was mixed with other buffers following the kit’s instructions and incubated at 37 °C for a certain time. The amplified products were purified and recovered with DNA purification kit and analyzed with agarose gel electrophoresis. The primer pairs with clear and bright amplification bands for *Burkholderia gladioli* but no specific amplification bands for other bacteria were preliminary selected. In addition, an initial CRISPR/Cas12a system consisted of 10μL RPA amplification product, 50 nM Cas12a, 50 nM crRNA, 200 nM ssDNA-FQ reporting probes, 0.8 U RNase inhibitor and NEBuffer 2.1, which was used to provide the fluorescence results. The primer pair that could specifically amplify *Burkholderia gladioli* and produce the highest fluorescence intensity was finally chosen. Next, the RPA system was optimized using the gDNA of *B. cocovenenans* and the best primer pair. The optimization conditions included forward/reverse primer concentration (0.05 μM, 0.1 μM, 0.15 μM, 0.2 μM, 0.25 μM), dNTPs concentration (1.6 mM, 1.8 mM, 2.0 mM, 2.2 mM, 2.4 mM), MgOAc concentration (14 mM, 17 mM, 20 mM, 23 mM, 26 mM), and amplification time (10 min, 15 min, 20 min, 25 min, 30 min). The optimal conditions were selected according to the fluorescence intensity generated by the initial CRISPR/Cas12a system mentioned above, and a nontarget control (NTC) using sterile water instead of gDNA was set for each optimized group. All tests were repeated three times, and the standard deviation was calculated and shown in the figures via the error bar.

### 2.4. Optimization of CRISPR/Cas12a Detection System

According to the sequence of the RPA amplification product, a 21 bp fragment with PAM site (TTTV-) was selected as the recognition target of crRNA, and a crRNA that specifically recognized the PRA amplification product was designed and synthesized. The crRNA sequence was: UAAUUUCUACUAAGUGUAGAUCAAGCAGGGGGUCGUCGGUUC (Figure 1). Next, the target fragments obtained with RPA amplification were purified for the CRISPR/Cas12a system optimization. First, the concentration of Cas12 enzyme was determined, and 60 nM, 80 nM, 100 nM, 120 nM, and 140 nM were selected as the optimized conditions for Cas12a enzyme concentration. The concentration optimization parameters of crRNA and ssDNA-FQ reporting probes were based on the ratio of the two to the optimized concentration of Cas12a enzyme. For crRNA concentration optimization, 0.5:1, 1:1, 1.5:1, 2:1, and 2.5:1 were selected as the conditions. For ssDNA-FQ reporting probes, 0.5:1, 1:1, 1.5:1, 2:1 and 2.5:1 were selected as the optimized conditions. The optimal conditions were selected according to the fluorescence intensity and cost. Nontarget control was set for each optimized group using sterile water instead of the target, and all experiments were repeated three times.

### 2.5. RPA-CRISPR/Cas12a Detection Procedure

Finally, the RPA-CRISPR/Cas12a detection method for *Burkholderia gladioli* was established, and specific steps are as follows. After DNA extraction, RPA amplification system with a total volume of 10 μL was added in test tube and covered with 15 μL of mineral oil to avoid aerosol pollution. Then, the CRISPR/Cas12a detection system was added on the inner face of the tube cover, which remained closed during the whole detection process. The tube was first cultured in a 37 °C constant temperature water bath for 20 min, then centrifuged by a handheld centrifuge for a few seconds, and it was cultured at 37 °C for another 10 min. The results can be immediately observed with naked eye under a portable blue light glue cutter. Fluorescence signals were collected using SpectraMax i3x (molecular devices, Sunnyvale, CA, USA) at an excitation wavelength of 492 nm and an emission wavelength of 518 nm.

### 2.6. Specificity and Sensitivity of RPA-CRISPR/Cas12a

In this study, three different strains of *Burkholderia gladioli* and five common foodborne pathogens, including *Salmonella*, *Vibrio parahemolyticus*, *Listeria monocytogenes*, *Bacillus cereus*, and *Staphylococcus aureus* were used to evaluate the specificity of the established RPA-CRISPR/Cas12a method.

The sensitivity of the method was determined from two levels: genomic DNA and the amount of bacteria. The detection limit of this method was determined as the lowest recognition concentration of gDNA or bacteria. The gDNA of *B. cocovenenans* was diluted with sterile water to 10^0^ ng/μL, 10^−1^ ng/μL, 10^−2^ ng/μL, 10^−3^ ng/μL, and 10^−4^ ng/μL, respectively. At the same time, the bacterial solution was also set to gradient concentration and used for detection together with the above diluted gDNA. Bacteria were cultured and counted according to the national standard of China (GB4789.2-2016) [44]. The bacterial solution in the logarithmic growth stage was serially diluted. A total of 100 μL of each dilution of the bacterial solution was plated on agar plates, and the colonies were counted. The concentration of bacterial solution was set up in six gradients, which were 10^6^–10^1^ CFU/mL. According to the significant difference of the fluorescence contrasted to the nontarget control, which was observed by naked eye, the detection limits of gDNA and bacterial concentration were determined. All experiments were repeated three times.

### 2.7. Real Food Sample Testing

In China, rice noodles, fresh white noodles, and glutinous rice flour are widely available and popular foods, but many of them are produced in processing places without food safety supervision, and the probability of contamination by *Burkholderia gladioli* is greatly increased [11]. The samples of rice noodles, fresh white noodles, and glutinous rice flour were first tested with qPCR according to the literature’s method [26], and the samples were confirmed to be “pollution-free”. QPCR reaction system is as follows: 12.5 μL 2 × TaqMan Fast qPCR Master Mix, 0.2 μM forward primer, 0.2 μM reverse primer, 0.2 μM probe, 1 μL gDNA, and 10 μL sterile water. The reaction process was as follows: pre-denaturation at 95 °C for 10 min, cycling, denaturation at 95 °C for 15 s, annealing at 65 °C for 30 s, collecting fluorescence signals, and repeating the cycle steps 40 times. The experiment set up positive control, negative control, and nontarget control at the same time. The template used for positive control was *B. cocovenenans* gDNA; the negative control was Salmonella gDNA, and the nontarget control was sterile water.

The rice noodle and fresh white noodle samples were directly homogenized on the ultra-clean workbench, and the glutinous rice flour was first added with sterile normal saline to make a 10-fold homogenate. Then, three samples were diluted 10-fold with sterile normal saline in the same way to make a suspension, which was allowed to stand for 2 min, and the supernatant was taken for later use. Next, 4.5 mL of the supernatant was mixed with 0.5 mL of the known concentration of *B. cocovenenan* to obtain 6 groups of samples with the concentration of *B. cocovenenan* of 10^6^ to 10^1^ CFU/mL. Finally, DNA was extracted using the kit extraction method. The established RPA-CRISPR/Cas12a method was used to test the detection limits of these samples. All experiments were repeated three times.

### 2.8. Method Validation

The qPCR method mentioned above was used to analyze *Burkholderia gladioli* at different concentrations in the three types of samples, and the results were verified and compared with those of the RPA-CRISPR/Cas12a method. The sample pretreatment method was consistent with the RPA-CRISPR/Cas12a method established in this study. The bacterial concentration gradients in the samples set for qPCR method was also the same with that of RPA-CRISPR/Cas12a method described above. *Burkholderia gladioli* was reported as “positive” when the final Ct value was less than or equal to 35. When the Ct value of the test result was greater than 35 and less than 40, the test was repeated. If the Ct value was still less than 40 and the curve had a significant logarithmic growth period, *Burkholderia gladioli* was also reported as “positive”; otherwise, *Burkholderia gladioli* was reported as “negative”. All experiments were repeated three times.

### 2.9. Statistical Analysis

Experimental data were statistically analyzed using IBM SPSS Statistics 26 software, and all results are presented as the mean ± SD of three independent experiments. One-way ANOVA was used to analyze the difference between the experimental group and the nontarget control group (NTC). The Tukey–Kramer multiple comparison method was used for post hoc test. The specificity and sensitivity of RPA-CRISPR/Cas12a method were judged according to whether the difference was significant. Significant values were indicated by * *p* < 0.05; ** *p* < 0.01; *** *p* < 0.001, and **** *p* < 0.0001.

## 3. Results and Discussion

### 3.1. Optimizing of RPA Amplification System

Nine pairs of primers were designed based on the highly conserved 16S-23S rRNA sequence in *Burkholderia gladioli*. Several pathogens that are easily contaminated in food, including *Salmonella*, *Vibrio parahemolyticus*, *Listeria monocytogenes*, *Bacillus cereus*, and *Staphylococcus aureus*, were tested to examine the specificity of the primers. At last, a pair of primers that could specifically amplify *Burkholderia gladioli* and produce the highest fluorescence intensity was screened out. The sequence is 5′-CCGTCTTGATAAGGCGGGGGTCGTTGGTTCGAAT-3′ for the forward primer and 5′-CGCCAATGACAAAGACTCGAGTCAACTGACCC-3′ for the reverse primer. Using this pair of primers, the RPA amplification system was optimized (Appendix A). In this study, the primer concentration recommended by the kit’s instructions was first tried to conduct the experiments, but the results were not good. Subsequently, smaller concentration ranges were selected as optimization parameters. The results showed that the primer concentration increased while the fluorescence intensity decreased within the optimization range, and the lowest primer concentration showed the best amplification efficiency. This phenomenon can be explained as the low concentrations of primers may reduce the amplification speed, but they may be beneficial to a longer amplicon and improve the real-time resolution. However, if the primer concentration was too low, the final amplification product would be insufficient and affect the subsequent experiments, so the optimal primer concentration was finally selected as 0.05 μM. The fluorescence intensity did not change significantly with the change in dNTPs concentration; therefore, the optimized concentration of dNTPs was set to 1.6 mM for cost saving. MgOAc mainly provides energy for the amplification process, and the concentration of MgOAc can affect the amplification efficiency of the enzyme. In this system, the fluorescence intensity was first increased and then decreased when the MgOAc concentration increased; thus, 20 mM was selected when the maximum fluorescence intensity was presented. Amplification time affects the amount of amplification product, but the fluorescence signal will gradually reach saturation with the increase in amplification time. With the increase in amplification time, the fluorescence intensity is the highest at 20 min, indicating that the amplification of *Burkholderia gladioli* can be completed within 20 min under the optimal reaction conditions. In addition, in order to comply with the concept of environmental protection and economy, the reaction system was reduced to a total volume of 10 μL, which is only one-fifth of the recommended reagent dosage of an RPA kit. Finally, the optimized system of RPA includes a 0.05 μM forward primer, 0.05 μM reverse primer, 1.6 mM dNTPs, 20 mM MgOAc, 2 μL DNA template, 5 μL 2 × Reaction Buffer, 1 μL 10 × Basic e-mix, and 0.5 μL 20 × Core Reaction Mix.

### 3.2. Establishment of RPA-CRISPR/Cas12a Method

In the CRISPR/Cas12a system, only when cas12a, crRNA, and target DNA form ternary complexes, Cas12a shows the trans-cleavage activity and can nonspecifically cut ssDNA-FQ reporting probes, thereby releasing a fluorescent signal [45]. Therefore, the concentration and ratio of cas12a, crRNA, and ssDNA-FQ reporting probes are the main factors affecting the detection performance. The purified RPA products were used to optimize the CRISPR/Cas12a system; the results are shown in Appendix A. First, a suitable concentration of Cas12a was preliminarily explored; the fluorescence intensity was improved with the increasing concentration of Cas12a enzyme. When the concentration of Cas12a was greater than 120 nM, the fluorescence intensity did not change significantly and had already met the requirements for rapid determination with the naked eye. Next, the concentrations of crRNA and ssDNA-FQ reporter probes were optimized in turn. With the increasing concentration ratio of crRNA/Cas12a, the fluorescence intensity first increased and then decreased, and the fluorescence intensity was the highest at 1:1. The results showed that excessive crRNA was not conducive to activating more Cas12a enzymes, so the concentration of crRNA was determined to be 120 nM. At the same time, the fluorescence intensity of the product showed an increasing trend with the increase in the ratio of ssDNA-FQ reporting probes/Cas12a. However, when the ratio was 2.5:1, the fluorescence intensity was sufficient to meet the demand, and the concentration of ssDNA-FQ reporting probes was determined to be 300 nM considering the economic cost. The final CRISPR/Cas12a system consisted of 120 nM Cas12a enzyme, 120 nM crRNA, 300 nM ssDNA-FQ reporting probes, 2.5 μL NEBuffer 2.1, 0.4 U RNase inhibitor, and refilled to a total volume of 15 μL with sterile water. The results showed that the CRISPR/Cas12a detection system could produce enough fluorescence for the naked eye observation within 10 min under the optimal reaction conditions.

The total volume of the combined RPA-CRISPR/Cas12a detection system is only 25 μL, including the 10 μL RPA solution and 15 μL CRISPR/Cas solution. For the convenience of the detection process, the CRISPR/Cas12a detection system was added to the cap of the test tube. The CRISPR/Cas12a and RPA solutions can be mixed by using a handheld centrifuge for just a few seconds, which ensures the continuous reaction and reduces the detection time to only 30 min. Compared with the method used by other researchers, which partially transfers the amplification products of RPA into the CRISPR/Cas detection system [46], this method is more efficient. At the same time, the operation of using 15 μL mineral oil to cover the liquid level of the RPA amplification reaction solution, with no need to open the tube cap during the whole reaction process, could effectively avoid false positive pollution. Compared with the practice of using a special capillary-based setup to control the reaction in an orderly way to avoid false positive pollution [47], this method is more convenient.

### 3.3. Method Performance Determination

The established RPA-CRISPR/Cas12a method theoretically provides dual specificity for the method due to the amplification reaction and the CRISPR recognition step. DNA extracted from *Burkholderia gladiolus* serotypes and several common food-borne pathogenic bacteria, including *Salmonella*, *Vibrio parahemolyticus*, *Listeria monocytogenes*, *Bacillus cereus*, and *Staphylococcus aureus*, were selected for specific reactions. As shown in Figure 2a, only the three *Burkholderia gladiolus* serotypes could activate Cas12a and produce visible fluorescence while no fluorescence was observed at all for other bacteria, indicating the high specificity of the RPA-CRISPR/Cas12a approach.

The sensitivity of the RPA-CRISPR/Cas12a method was tested at both the gDNA and bacteria level. As shown in Figure 2b, it was still visible to the naked eye at 10^−3^ ng/μL (equal 10^0^ pg/μL) gDNA concentration and obviously different from the nontarget control. Meanwhile, *B. cocovenenans* colonies were cultured and diluted to different concentrations in series, and the extracted DNA was used for RPA-CRISPR/Cas12a detection. According to the results shown in Figure 2c, the obvious detection limit was 10^1^ CFU/mL for bacterial fluid. Compared with the reported qPCR method [26], the detection limit of qPCR for *Burkholderia gladioli* gDNA was 250 fg/μL (equal 0.25 pg/μL), and for the bacterial solution, it was 10^2^ CFU/mL. In comparison, the detection limit of this method was slightly worse than that of the qPCR method, but the detection limit of the actual bacteria was better than that of the qPCR method. This may be due to the lower tolerance of qPCR to inhibitors than RPA, resulting in the two methods showing different result trends when detecting bacterial liquid samples compared with purified DNA [48]. This established RPA-CRISPR/Cas12a method does not require large instruments, is easier, and is suitable for rapid detection in the field.

### 3.4. Application in Real Food Samples

In this study, homogenized rice noodles, fresh white noodles, and glutinous rice flour samples were first proven free from contamination by *Burkholderia gladioli*, and then, they were spiked with *B. cocovenenans*. The final bacteria concentration in the sample ranges from 10^0^ CFU/mL to 10^6^ CFU/mL. In the rice noodles, fresh white noodles, and glutinous rice flour samples, the RPA-CRISPR/Cas12a assay showed a detection limit of 10^1^ CFU/mL, 10^2^ CFU/mL, and 10^2^ CFU/mL against *Burkholdeia gladioli*, respectively (Figure 3). The detection limit of rice noodle samples is the same as that of the bacterial solution while the detection limit of fresh white noodles and glutinous rice flour samples is an order of magnitude higher than that of the bacterial solution. The reason may be attributed to different water content of different samples. The fresh white noodles and the dry glutinous rice flour contains less moisture content than that of the rice noodles; therefore, they may offer more matrix effects, which disturb the detection, such as during the DNA extraction process. The presence of a sample matrix may block the DNA adsorption membrane in the kit, resulting in a decrease in the amount of extracted DNA, and the matrix still exists in the final extract to be tested, which may interfere with the amplification efficiency and detection effect of the system and may eventually lead to a higher detection limit. In addition, it was very difficult to extract DNA from glutinous rice flour due to the high sample viscosity. The whole detection process of real food samples, including sample processing, DNA extraction, and the detection step, takes no more than 1 h and with no need of a sample enrichment.

### 3.5. Method Validation

The accuracy of the RPA-CRISPR/Cas12a method was verified by an existing qPCR method in the three kinds of samples; as shown in Figure 4, the detection results were represented by the CT value. The detection limit of *Burkholderia gladioli* in rice noodles, fresh white noodles, and glutinous rice flour obtained with qPCR was 10^1^ CFU/mL, 10^2^ CFU/mL, and 10^3^ CFU/mL, respectively. The results obtained with qPCR showed a consistent trend with that of the RPA-CRISPR/Cas12a method in all three kinds of samples. In fact, the RPA-CRISPR/Cas12a method is even more sensitive to the existing qPCR methods. This may be caused by the different tolerance of qPCR and RPA to the inhibitory effects brought by the sample matrix [49]. The above results indicate that the RPA-CRISPR/Cas12a method had good accuracy and sensitivity.

## 4. Conclusions

This study established a specific RPA-CRISPR/Cas12a rapid method to detect the food-borne pathogen *Burkholderia gladioli*. This method has high sensitivity and good specificity that can precisely detect *Burkholderia gladioli* without a cross-reaction with other common pathogens. The detection limit of gDNA was 10^−3^ ng/μL, and the limit of detection of the bacterial liquid was as low as 10^1^ CFU/mL, which was better than that of the standard qPCR method. The method was then applied in food samples, and the detection limits of rice noodles, fresh white noodles, and glutinous rice flour were 10^1^ CFU/mL, 10^2^ CFU/mL, and 10^2^ CFU/mL, respectively, with no need of a sample enrichment. The detection step only took 30 min, and the whole process of food sample detection with the addition of sample processing and DNA extraction takes no more than 1 h. There are few studies on the rapid detection method of *Burkholderia gladioli*, and most of the food contaminated samples are rice noodle products, which also have some difficulties in pre-processing. However, this method has high sensitivity, which can overcome the complex problem of DNA extraction from rice noodle products. Moreover, this rapid method can be carried out without the assistance of any large instruments, but it achieves even better sensitivity than qPCR, so it has good prospects for the efficient and rapid field detection of *Burkholderia gladioli*.

## Figures and Tables

**Figure 1 foods-12-01760-f001:**
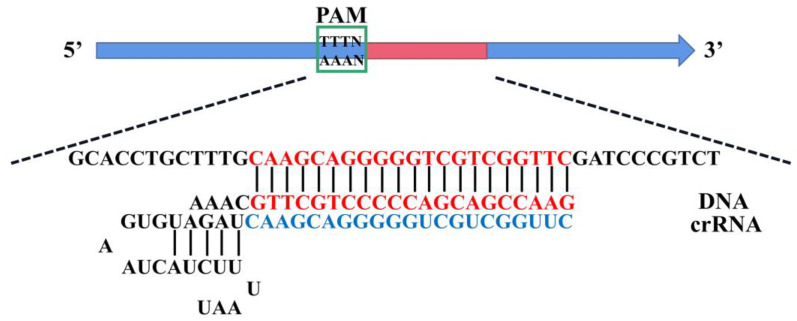
Schematic representation of double-stranded DNA (dsDNA) targets detected by complexes of Cas12a and CRISPR-derived RNA (crRNA).

**Figure 2 foods-12-01760-f002:**
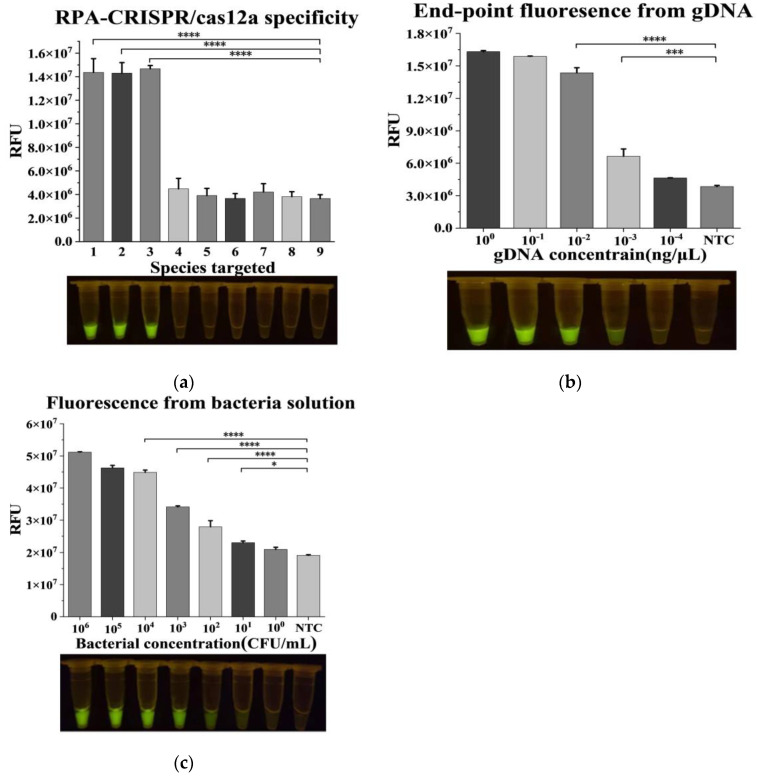
Establishment and evaluation of Recombinase polymerase amplification (RPA)-CRISPR/Cas12a method. (**a**) Specificity analysis. Among them, the pathogenic bacteria were as follows: (1) *Burkholderia Gladiolus* (CICC10574), (2) *Burkholderia Gladiolus* (ATCC10248), (3) *Burkholderia Gladiolus* Toxigenic strains, (4) *Salmonella*, (5) *Vibrio parahaemolyticus*, (6) *Listeria monocytogenes*, (7) *Bacillus cereus*, (8) *Staphylococcus aureus*, and (9) nontarget control (use sterile water). (**b**) Sensitivity analysis at genomic DNA (gDNA) level. (**c**) Sensitivity analysis for bacterial quantity. n = 3 biological replicates, error bars represent the standard deviation of three replicates. One-way ANOVA; Tukey–Kramer was used for post hoc comparisons; * *p* < 0.05, *** *p* < 0.001, and **** *p* < 0.0001. NTC: nontarget control. RFU: relative fluorescence unit.

**Figure 3 foods-12-01760-f003:**
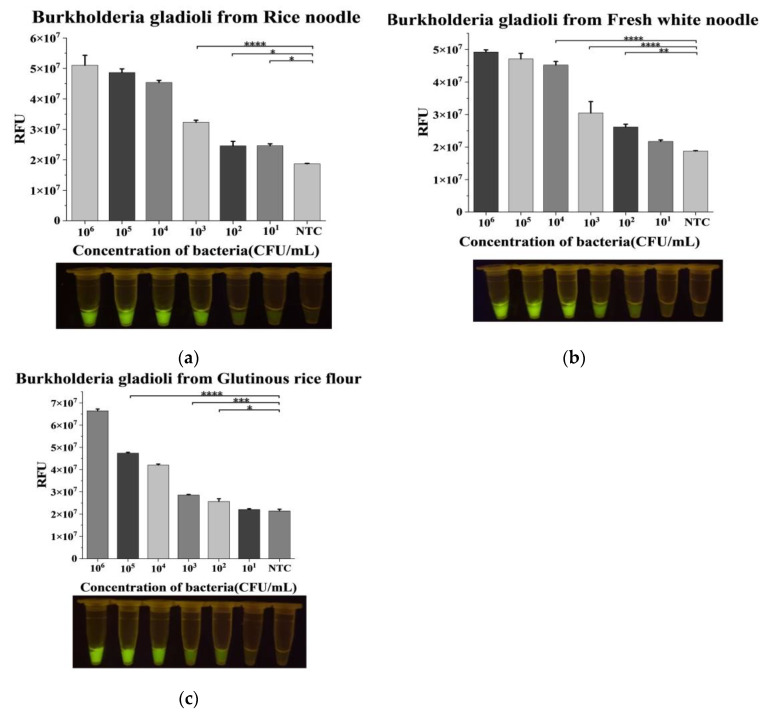
Application of RPA-CRISPR/Cas12a system in real food matrix. (**a**) Feasibility and sensitivity analysis of contaminated rice noodles. (**b**) Feasibility and sensitivity analysis of contaminated fresh white noodles. (**c**) Feasibility and sensitivity analysis of contaminated glutinous rice flour. n = 3 biological replicates, error bars represent the standard deviation of three replicates. One-way ANOVA; Tukey–Kramer was used for post hoc comparisons; * *p* < 0.05, ** *p* < 0.01, *** *p* < 0.001, and **** *p* < 0.0001. NTC: nontarget control. RFU: relative fluorescence unit.

**Figure 4 foods-12-01760-f004:**
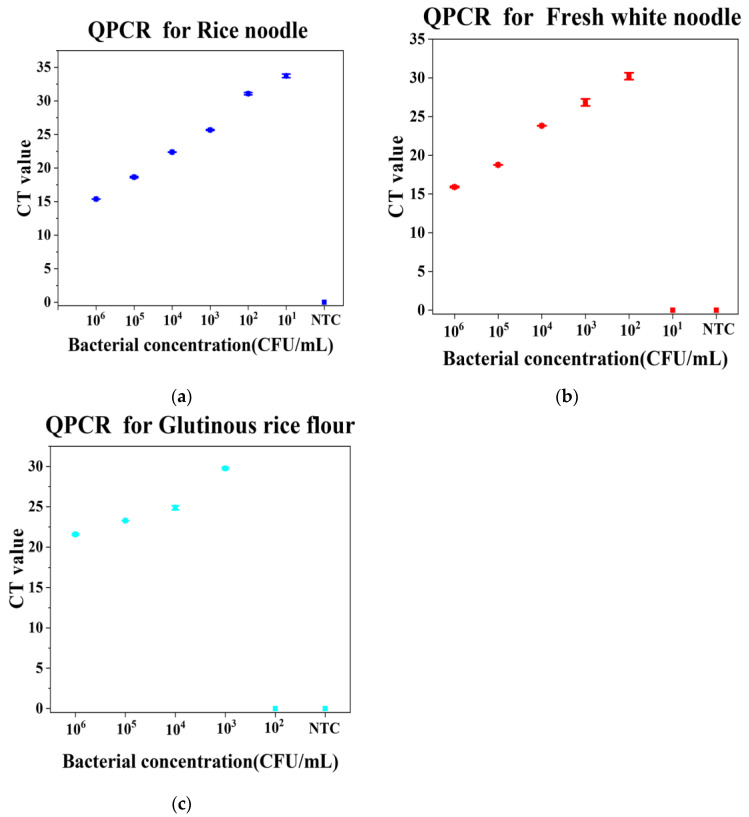
Method confirmation with QPCR. (**a**) Rice noodle sample. (**b**) Fresh white noodle sample. (**c**) Glutinous rice flour sample. n = 3 biological replicates, error bars represent the standard deviation of three replicates. NTC: nontarget control. CT: Cycle Threshold.

## Data Availability

Data is contained within the article (or Appendix A).

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
