# Peer review of "Rapid and Simple Detection of *Burkholderia gladioli* in Food Matrices Using RPA-CRISPR/Cas12a Method"

_foods, 2023, doi:10.3390/foods12091760_

Round 1
Reviewer 1 Report
The manuscript entitled “Rapid and simple detection of Burkholderia gladioli in food matrics using RPA-CRISPR/Cas12a method” addresses the use of quick and easy visual detection of Burkholderia gladioli, a serious threat to human health, in the field detection. This was accomplished by using an optimized RPA-CRISPR/Cas12a assay. The assay revealed a detection limit of 10-3 ng/μL and 101 CFU/mL for genomic DNA and bacterial quantity, respectively. For the current study results, the authors concluded that the established RPA-CRISPR /Cas12a method is simpler and more sensitive compared with the qPCR method. The current findings are interesting.
Comments:
1) The title of the current study reads “Rapid and simple detection of Burkholderia gladioli in food matrics using RPA-CRISPR/Cas12a method”. What do the authors specifically mean by “food matrics”? The authors are advised to use the term “Food matrices” instead.
2) In the supplementary material, the authors are advised to add in Table S1 (Nine sets of primer sequences and size) the gene accession number for each primer set.
3) In the material and methods section, the authors are advised to add a separate section for the “statistical analysis”. In that section, the authors should describe the name of the used software to run statistical analysis, and how the data were presented as Mean ± SD or SE. Did the authors check data normality and homogeneity using Shapiro-Wilk test, for example, before proceeding to statistical analysis? What is the used p-value?
4) As described in the legend of Figure 2, the authors state “two-tailed Student's t-test; *P < 0.05, **P < 0.01, 322***P < 0.001, and ****P < 0.0001”. However, the use of a two-tailed Student's t-test herein is not an appropriate statistical test since the authors are comparing several treatments/concentrations. Ideally, the authors should be analyzed by one-way ANOVA followed by a post-hoc test e.g., Tukey-Kramer. The authors are advised to do the statistical analysis as described above. Authors are advised to address this point and add the answers in the statistical analysis section and the relevant figure legends.
5) In the material and methods section, the qPCR is missing biological (how many samples were used per group) and technical repeat information (whether each sample was repeated during the assay).
6) In qPCR, did the authors check the DNA quality with A260/280? Please, add these data in the relevant section in material and methods.
7) To make all figure legends stand-alone, authors are advised to add the full name of the used abbreviations at the end of each legend including RPA-CRISPR, RFU, gDNA, etc.
8) Careful revision of the reference list should be performed. For example, reference no. 25 is written in ALL CAPS format. Please, revise the entire reference list.
9) The authors are advised to stick to the journal guidelines for writing the reference list. Herein, all the journal names are written as full names. However, they should be written in an abbreviated form. In writing the page range, please avoid using “p.”. Here is an example, that you may need to follow:
“Jin, Z.; Lan, Y.; Ohm, J.-B.; Gillespie, J.; Schwarz, P.; Chen, B. Physicochemical Composition, Fermentable Sugars, Free Amino Acids, Phenolics, and Minerals in Brewers’ Spent Grains Obtained from Craft Brewing Operations. J. Cereal Sci. 2022, 104, 103413”.
10) The current manuscript needs to be carefully checked by a native English speaker for grammar, syntax, and typos. Some typos/syntax errors are present in the manuscript that need to be addressed, for example:
A) In lines 16-17, the authors state that “The optimized RPA-CRISPR/Cas12a assay was able to specifically and stably detect Burkholderia gladioli at a constant 37℃ without the assistant of large equipment”.
Please, consider correcting the above statement to “The optimized RPA-CRISPR/Cas12a assay was able to specifically and stably detect Burkholderia gladioli at a constant 37℃ without the assistance of large equipment”.
A) In lines 22-23, the authors state that “Compare with qPCR method, the established RPA-CRISPR /Cas12a method was much simpler and even more sensitive”.
Please, consider correcting the above statement to “Compared with the qPCR method, the established RPA-CRISPR /Cas12a method was simpler and even more sensitive”.
The current manuscript needs moderate editing in the English language. It needs to be checked by a native English speaker for grammar, syntax, and typos.
Reviewer 2 Report
Zheng and co-authors designed of RPA-CRISP-Cas12 based detection system for detection of the rice pathogen Burkholderia gladioli. The system provides highly sensitive and specific detection of this pathogen in low equipment facility within 1h. Undoubtedly the research described in the paper has practical significance and can be used for development of a protocol for manufacturing a commercial test system. However, I would like the following notion will be taken into consideration and applied to the manuscript:
66-67 Class II Cas proteins also demonstrate powerful ability in rapid and ultrasensitive detection of nucleic acid demonstrate powerful ability in rapid and ultrasensitive detection of nucleic acid
Class II Cas proteins themselves have quite high sensitivity but not ultrasensitive. Combination of the proteins with amplification approaches provides ultrasensitivity (look J. S. Gootenberg et al., Science 10.1126/science.aam9321 (2017). – Fig 1C). I rather consider ultra-high specificity as the main advantage of biosensors.
98: The crRNA and ssDNA-FQ reporting probes were synthesized by
What dye and quencher were used?
114-115: The highly conserved 16S-23S rRNA sequence in Burkholderia gladioli was selected as the target for the design of specific RPA
How many copies of the intergenic spacer are in genome B.gladioli?
176: with sterile water to 10 0 ng/μL, 10 -1 ng/μL, 10 -2 ng/μL, 10 -3 ng/μL, and 10 -4 ng/μL,
Could you provide molar concentration or copies/mcL?
283-284: The total volume of the combined RPA-CRISPR/Cas12a detection system is only 25 μL including 10 μL RPA solution and 15 μL CRISPR/Cas solution.
RPA mix is almost 50% of the reaction mix. Could RPA compounds impact Cas12 activity? Have you tried testing the lower portion of RPA solution in the final mix? Can less portion of RPA mix in the final mix demonstrate better performance?
306-307: it was still visible to the naked eye at 10-3 ng/μL (equal 100 pg/μL)
Please provide sensitivity in molar or copy concentration as well.
309-310: According to the results shown in Fig. 2c, the obvious detection limit was 10 1 CFU/mL for bacterial fluid.
Please assess number of DNA targets in the sample corresponding 10 CFU/mL using number of target genes per cell. And compare to sensitivity of purified gDNA.
313-314: In comparison, the detection limit of this method was slightly worse than that of qPCR method, but the detection limit of actual bacteria was better than that of qPCR method.
PCR is less tolerant to inhibitors than RPA ( Wilson IG. . Appl Environ Microbiol. 1997 doi: 10.1128/aem.63.10.3741-3751.1997). So it can be a possible explanation of different trends for purified gDNA and bacterial samples.
360-361: In fact, the RPA-CRISPR/Cas12a method is even more sensitive to the existing qPCR methods.
I think it might be caused by different tolerance of PCA and RPA to inhibition by plant matrix (e.g. see review Ivanov AV, et.all.. Plants (Basel). doi: 10.3390/plants10112424)
Before, I should state that English is not my native language. I think the Quality of English Language is quite good and does not require extensive proofreading.
